# Etonogestrel-releasing subdermal contraceptive implant: Budget impact analysis based on the Brazilian private healthcare system

**Agnaldo Lopes da Silva Filho**[1], **Ricardo Luis Pereira Bueno**[2,3], **Yohanna Ramires**[3,4],
**Lara Marina Cruz Lino**[3,5]*

1 Medicine Faculty, Gynecology Department, Federal University of Minas Gerais, Belo Horizonte, Minas Gerais, Brazil, 2 Business Graduate Program of University 9 of July (PPGA-UNINOVE), São Paulo, Brazil, 3 Organon & Co., São Paulo, Brazil, 4 Postgraduate Program in Pharmaceutical Science, Federal University of Paraná, Curiba, Paraná, Brazil, 5 Master's Business Administration in Health Program, Pontifícia Universidade Católica Do Rio Grande Do Sul, Porto Alegre, Rio Grande do Sul, Brazil

* lara.lino@organon.com

**Data Availability Statement:** All data are available from the below links: Data base: site: ans.gov.br/images/stories/Materiais_para_pesquisa/Perfil_setor/sala-de-situacao.html and BI where the

## Abstract

High rates of unplanned pregnancies persist despite pharmacological developments and advancements in contraceptive methods. Here, we demonstrate that the etonogestrel-releasing subdermal contraceptive implant (IMP-ETN) may be an appropriate and cost-effective alternative to levonorgestrel-releasing intrauterine systems (LNG-IUSs) for women in Brazil. For our pharmacoeconomic analysis, we reviewed the literature on IMP-ETN regarding its acceptance, eligibility criteria, choice, relations with age, adverse events and, finally, the unmet need in the fee-for-service private healthcare sector. We considered qualitative observations in combination with quantitative analysis and performed a deterministic sensitivity analysis to investigate whether this technology can be self-sustainable over a period of five years. The target population for this analysis comprised 158,696 women. Compared with the continued use of LNG-IUSs, adopting the IMP-ETN can result in a cost avoidance of $ 7.640.804,02 in the first year and $ 82,455,254.43 in five years. Disseminating information among physicians will promote this change and strengthen the potential cost avoided by private health system payers. These savings can be used to improve other healthcare programs and strategies. Moreover, the principles of care can be promoted by improving and adapting healthcare systems and expanding treatment and follow-up strategies. This would also provide support to women's reproductive rights and improve their quality of life. Our results suggest that the IMP-ETN has a favorable cost-effectiveness profile. Given all its advantages and negative incremental cost impact over a period of five years, the IMP-ETN may be a more favorable alternative to LNG-IUSs. Therefore, it should be offered to beneficiaries with a private healthcare plan. This analysis overcomes previous barriers to the use of cost-benefit models, and our results may help balance decision-making by policymakers, technical consultants, and researchers.

women by age were selected: ans.gov.br/images/
stories/Materiais_para_pesquisa/Perfil_setor/sala-
de-situacao.html and Microsoft Power BI.

**Funding:** The authors received no specific funding
for this work. ☐ Initials of the authors who
received each award - THERE WERE NO AWARD
☐ Grant numbers awarded to each author -
THERE WERE NO GRANT ☐ The full name of each
funder - ORGANON BRASIL ☐ URL of each funder
website - ORGANON.COM.BR ☐ Did the sponsors
or funders play any role in the study design, data
collection and analysis, decision to publish, or
preparation of the manuscript? LARA LINO,
YOHANNA RAMIRES, RICARDO BUENO ALL OF
THE ABOVE - WE THREE DECIDED TO PUBLISH,
COLLECTED THE DATA, WROTE AND INVITED
AGNALDO LOPES TO REVISE AND INPUTS.

**Competing interests:** Competing interest: I have
read the journal's policy and the authors of this
manuscript have the following competing interests:
Ricardo Luis Pereira Bueno, Yohanna Ramirez and
Lara Marina Cruz Lino are employees of Organon
Brasil [a commercial funder of this study]. This
does not alter our adherence to PLOS ONE policies
on sharing data and materials. Agnaldo Lopes da
Silva Filho declares no competing interests.

## Introduction

Unintended pregnancies raise several questions for clinicians and policymakers regarding the root of the problem and strategies for dealing with the consequences. High rates of unplanned and unintended pregnancies persist despite pharmacological developments in contraceptive methods. This indicates that access to contraceptives and freedom of choice may be more important in reducing the number of unintended pregnancies than the development of state-of-the-art progestins. Long-acting reversible contraceptives (LARCs) are the most recommended method of contraception because they do not require any interference from the recipient after insertion; that is, the user cannot alter or affect the efficacy of LARCs [1]. The availability and diversity of contraceptives determine their usage and ability to meet the needs of women (depending on their comorbidities, social status, and stage of life). As such, improving access to different contraceptive methods can have a positive impact on maternal and child health.

Reproductive health is a priority in primary healthcare and is considered a fundamental human right by the United Nations (UN), which encourages the dissemination of information and access to contraceptive methods and pregnancy diagnosis for men and women [2]. However, the contraceptive method of choice must meet the clinical, psychological, cultural, and religious needs of its users [3]. Overall, LARCs have been shown to be the most effective method of birth control. In controlled clinical research settings and in the real world, LARCs are associated with a low number of contraceptive failures per 100 women/year (defined as the Pearl index). Perfect contraceptive usage is indicated by a Pearl index of 0–0.6, and values of 0.05–0.8 have been reported for real-world usage (depending on the type of LARC). In terms of absolute numbers, implants have been shown to outperform other contraceptive methods. However, LARCs are not commonly used in practice [4,5]. Etonogestrel-releasing subdermal contraceptive implants have a lower failure rate than hormonal contraceptives or copper intrauterine devices (IUDs), resulting in a lower incidence and probability of unintended pregnancy [6,7]. These implants demonstrate a higher efficacy than IUDs in young women, as indicated by their high rate of pregnancy prevention [5].

In addition to their contraceptive effects, LARCs trigger positive reactions in many women. A previous study analyzing pharmacoeconomics records identified 15 relevant studies; of these, 4 directly investigated contraceptive methods being used for contraception, whereas all others focused on contraceptives being used to treat clinical conditions (such as uterine fibroids, endometriosis, and dysmenorrhea) [8]. Chen et al. evaluated a proof-of-concept in users of combined hormonal contraceptives (CHCs) who agreed to use a subdermal implant of etonogestrel (IMP-ETN) as a "back-up" contraceptive [9]. The implant showed high efficiency as a LARC, and the authors found that nearly 40% of the participants preferred to continue using only the implant and stop CHC usage. Outcomes like these indicate that the IMP-ETN is valued by its users in terms of tolerability and satisfaction. However, given the small sample size, further studies are needed to validate their findings.

In this context, it is important to analyze LARCs in the pharmacoeconomic context of the Brazilian Private Healthcare System. In Brazil, healthcare is provided either by the Unified Health System (*Sistema Único de Saúde*–SUS)—which ensures universal access to healthcare and is exclusively funded by public resources—or by private healthcare providers (where healthcare plans are financed by families and/or employers through direct disbursement) [10]. In the private healthcare system, payments are predominantly based on the fee-for-service model, which is expensive. Therefore, it is important to identify more effective ways of allocating funds to ensure the sustainability of the private healthcare system [10,11].

Currently, only hormonal intrauterine devices (h-IUDs)—also known as intrauterine systems (IUSs)—are available in the Brazilian private healthcare system. This exclusivity provides

IUD manufacturers with a competitive advantage in the form of a temporary monopoly and possible value maximization. While it is true that the intrauterine system of levonorgestrel is a reliable contraception method for women who opt for private healthcare services, an equally effective non-uterine option should also be offered [12]. The authors evaluated that the provision of the levonorgestrel-releasing intrauterine system (LNG-IUS) at no cost for low-income Brazilian women averted unintended pregnancies, maternal and child mortality, and abortions. They concluded that the data of disability adjusted life years (DALY) averted found from LNG-IUS could equally apply to the copper intrauterine device or contraceptive implants [12].

The entry of new competitors into the market, as in any other economic model, improves the sustainability of the system by forcing manufacturers to adopt more competitive prices, simplify production processes, learn from competitors' experiences, and enhance products, resulting in improved expected outcomes [13]. A major challenge today is access to quality data for reliable economic analysis and the definition of cost measurement standards. Another paradigm is the migration from payment for volume towards payment for performance. In this sense, the concept of health as a volumetric and quantitative idea needs to shift toward an individualized and qualitative experience for each patient. An economy that places value at the center of health services is fundamental, as value must take into account all stakeholders, including patients, doctors, as well as administrative departments of hospitals. We also understand the importance of sustainability of the private fee-for-service model system. However, it is essential to maintain an open perspective and understand the clients' overall view and expectations in order to achieve service excellence.

Few studies have assessed healthcare technologies related to contraceptives and their impact on family planning in low-income countries. Moreover, the assessments from developed countries are not feasible for implementation in low-income countries because differences in socio-economic structures, cultural conditions, and financial models can lead to erroneous conclusions [14].

## Main evidence

The IMP-ETN contains c3-keto-desogestrel (3-KDSG), and the device consists of a single rod containing 68 mg of etonogestrel. For correct usage, it is recommended that the implant be placed under the skin of the upper non-dominant arm. The implant remains active and effective for up to three years and releases progestin in minimal concentrations, which helps maintain safety while preventing unwanted pregnancy [7]. The mechanism of action of the implant is twofold. First, it inhibits ovulation by preventing the mid-cycle peak of luteinizing hormone [15]. Initially, the hormone suppresses follicular development and estradiol production. Ovarian activity increases gradually after six months, at which point follicle stimulating hormone and estradiol reach their normal physiological levels. Therefore, the implant blocks ovarian function and inhibits almost all cycles in the short term by preventing LH release and ovulation [1,16]. It also thickens the cervical mucus, which reduces the entry of spermatozoa [15]. The endometrium is also modified by the progestin, which prevents the implantation of potential fertilized eggs [7].

Data from the Contraceptive CHOICE Project showed that adolescents aged 14–19 years reported high continuation rates (82% at 12 months) for the implant [17] The World Health Organization suggests that subject to the eligibility criteria, the majority of women can safely use these implants [18]; however, women who are hypersensitive to barium or to other components of the implant are exempted from this recommendation [7].

As with any other method—whether hormonal or not—side effects can occur with implants as well. The side effects of implant are similar to those associated with the use of progestins.

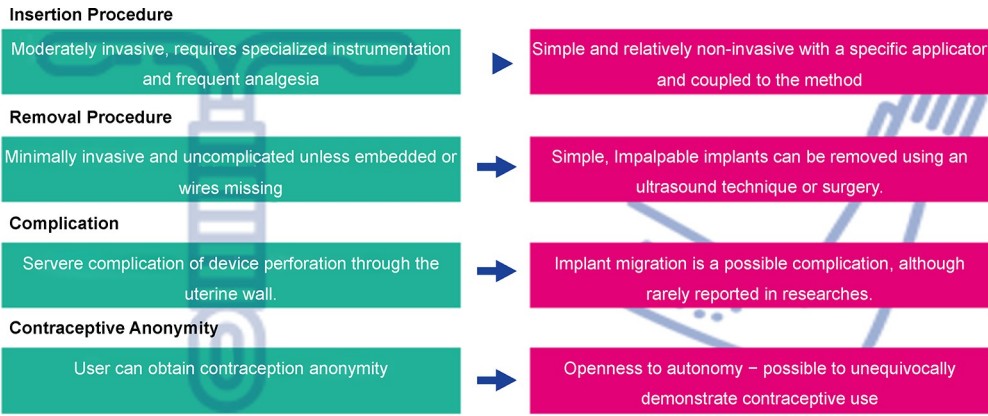

**Fig 1. Differences between procedures and user autonomy.** Created by the authors—Based on [23].

The most frequent side effect in this case is unscheduled vaginal bleeding. For every 100 pg/mL increase in serum etonogestrel concentration, contraceptive implant users in this study had 1.6 times the odds of reporting abnormal bleeding and 2.3 times the odds of having received a prescription as treatment for bothersome bleeding [19]. Acne, weight gain, and reduced libido are reported less frequently [7]. All these side effects can be managed with accessible medications and practices, which can help maintain the successful functioning of the implant.

No previous studies have reported the expulsion rates for this method. The insertion is subdermal and is relatively safe and superficial [18]. A monitoring program collected real-world data regarding this technique based on >26,000 insertions, and the findings highlighted the safety of the procedure. The total number of reported adverse events related to insertion, localization, or removal of the etonogestrel implant was extremely low [20]. Moreover, the NORA study concluded that adverse events associated with the insertion, localization, and removal of the Nexplanon contraceptive implant were rare and that their clinical consequences were generally not suggestive of serious injury [21]. Lazorwitz et al. conducted an exploratory study and found no significant effect of diet or exercise on steady-state pharmacokinetics among users of contraceptive implant [22,23]. Another study observed that implant users were significantly more likely to continue this method than those who selected other methods ($p<0.001$) [24].

The insertion and removal procedures of non-palpable implants deserve some discussion. Insertions that are too deep (e.g., under the brachial fascia or in subcutaneous tissue) can complicate removal, although these reports are highly debated. However, deeply inserted contraceptive implants can be removed under continuous ultrasound guidance alone, and this method is minimally invasive, feasible, effective, and safe [22]. Nonetheless, the better the insertion of the implant, the easier and quicker its removal (Fig 1).

Pires et al. provided some recommendations and suggested reasons for discontinuing the use of the LNG-IUS system [25]. The published survey also showed that only 32.7% of IUDs are used for contraceptive purposes. Although IUDs are a widely used LARC option with a high satisfaction rate, their side effects and complications must be presented to women to facilitate informed decision making (Fig 2).

According to Ferreira et al., 2014, the fear of becoming pregnant alone is responsible for women changing their contraceptive method in 59% of cases. One year after placement, the primary reasons for discontinuation with the LARC method were expulsion for both the copper IUD and the LNG-IUS, and a desire to undergo surgical sterilization in case of the ENG-releasing implant. The continuation rate was around 95% women/year. The most common

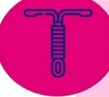

| • Pelvic pain | • Bleeding of a benign nature |
| • Abnormal uterine bleeding | • Several treatment options are available and should be offered prior to discontinuation. |
| • Post-coital bleeding | |
| • Menstruation disorders | |
| • Inflammation | • The optimal treatment strategy has not been elucidated, so the choice is based on the clinical setting and patient preference. |
| • Infection | |
| • Uterine perforation | |
| • Mechanical complications of the device | |
| • Expulsion | |
| • Ectopic and topical pregnancy | |

**Fig 2. Side effects and complications of LNG-IUS vs. IMP-ENG.** Created by the authors—Based on [26,27].

method for switching to an LARC was the combined oral contraceptive pill. However, only 2.2% opted for the implant. The authors state that "the implant is not commonly available in the public sector in Brazil, which prevents the public health authorities from acquiring it. Consequently, the implant was not always available at their clinic. Women who choose the implant as a method often have to purchase it privately, which constitutes a great barrier to the underprivileged segment of the population" [28]. As such, healthcare providers in Brazil should consider the value of proper information dissemination for their clients (Figs 3–5).

A

| Implant | LNG-SIU |
|---|---|
| • Greater adherence the lower the age group, reflecting a greater result as a contraceptive method per se due to behavior | • Breakthrough bleeding episodes and days are more frequent (pharmacological physiology) |
| • Option for need for anovulation - Package insert | • There are studies of efficacy in extended use |
| • Preferable when it involves uterine situations (postpartum, post-abortion, malformations) | • There may be a relationship with breast CA |
| • There are studies of efficacy in extended use | • It has an acceptable but regressive therapeutic effect |
| • Time does not compromise [ ] and therapeutic effect | |

B

**Trend by age and parity**

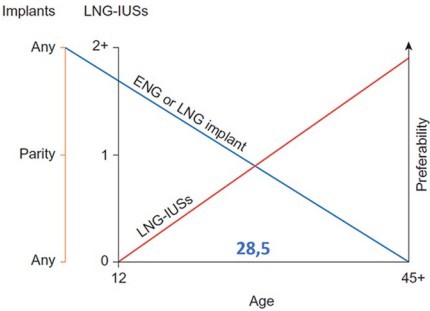

**Fig 3. Comparison of pharmacological effects and preference trends for implants vs. LNG-IUS.** Created by the authors—Based on A) Pharmacological effects of implants vs. LNG-IUS [7,29,30]. B) Idealized graph showing user age and parity relative to the probability of subdermal implant or LNG-IUS being preferable [29]. Abbreviations: IUD, intrauterine device; LNG-IUD, levonorgestrel-releasing IUD; ENG, etonogestrel implant.

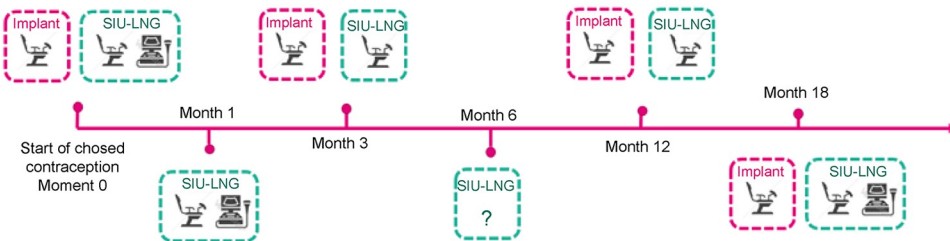

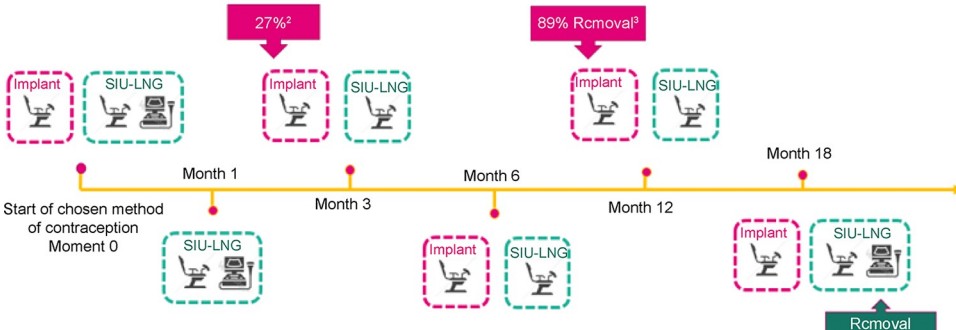

**Fig 4. A patient's journey from device insertion to different scenarios based on user satisfaction.** Created by the authors—Based on [31,32].

The CHOICE project, an emblematic study on contraceptive choice, revealed that well-informed women tend to prefer long-term and more efficient methods. These methods also offer greater convenience. The study found that 46% of women chose the levonorgestrel intra-uterine system, 12% chose the copper IUD, and 17% chose the subdermal implant. The data were collected by providing the option for same-day insertion, which resulted in increased adherence, satisfaction, loyalty to healthcare providers, and convenience [17].

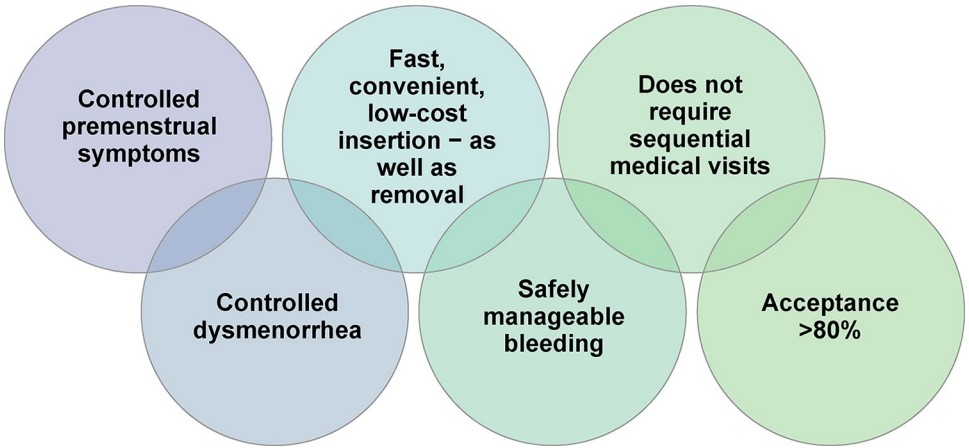

**Fig 5. Added value provided by subdermal contraceptive implants.** Created by the authors of this study.

Using Fig 4 as a base scenario, we need to highlight the factors that affect women's decision-making when choosing a contraceptive method. D'Souza (2022) conducted a review and synthesis and clearly illustrated that the use of contraceptives depends on the woman's environment and her perceptions and beliefs about motherhood. The choice is also influenced by the risks and threats to health, as well as the potential benefits that come with it. Expanding on the range of influential factors, women's choices are also affected by their partners, family, colleagues, healthcare providers, and society as a whole. Their knowledge and decision-making capacity play a role as well. The selection of a contraceptive method is also influenced by the characteristics of the methods themselves, including invasiveness, discretion, and impact on intimacy. Furthermore, the characteristics of the services, such as accuracy of information provided, advice, convenience, confidentiality, costs, and the attitudes and behaviors of health professionals also come into play [33].

In this study, we evaluate the economical and qualitative predicates of women in Brazil and demonstrate that etonogestrel-releasing subdermal contraceptive implants satisfy these criteria. Regardless of whether these products are distributed through private or publicly funded means, our conclusions are supported by the premise of women's right to reproductive planning.

## Materials and methods

For our pharmacoeconomic analysis, we reviewed previous literature to understand how information about IMP-ETN can impact its acceptance in the private healthcare sector. We combined qualitative observations with quantitative analysis to investigate whether IMP-ETN technology can be self-sustainable in the Brazilian private healthcare system.

### Pharmacoeconomic analysis

We performed pharmacoeconomic analysis using epidemiological estimates available in the literature and in publicly available data (number of beneficiaries, number of beneficiaries were female divided by age and number of LNG-IUS insertions) provided by the ANS (the regulatory agency of Brazilian private healthcare) through *Caderno 2.0* and from different government databases audited by the Integrated Inspection System (SIF) [34]. The target population included healthcare beneficiaries aged 20–44 years (those of childbearing age who fit the description in the medication package insert), divided into five-year age groups. To this number, we applied the percentages published by Lago et al., who researched contraceptive practices in a group of 4,000 Brazilian women and found that 84.8% of the women used contraception [35]. To estimate the number of women who could be targeted with this new LARC technology, we used the number of h-IUD insertions performed by healthcare providers in 2020 to calculate the percentage of women who used LARCs. This group of women was considered the target population for the present study [34]. To compare the effects of LARCs with those of hormonal contraceptives, we used the levonorgestrel (LNG) intrauterine system (LNG-IUS) as a comparator. Similar to an etonogestrel implant (ENG), the LNG-IUS system functions by releasing pharmacologically active progesterone molecules in users [29].

We estimated the target population for our analysis over a time horizon of five years. This period incorporated the maximum effective duration of the contraceptive methods and assumed an annual growth of 7% (similar to the increase considered for IUDs), according to data observed and documented by ANS [36]. Projected costs within the established time horizon were estimated from the perspective of private healthcare providers. The cost of each type of contraception was calculated based on the use of mandatory Brazilian professional spread sheets, to guarantee comparability throughout the national territory, according to the values

defined by the *Classificação Brasileira Hierarquizada de Procedimentos Médicos* (CBHPM; the Brazilian Hierarchical Classification of Medical Procedures, a referenced tool adopted as a minimum and ethical standard for remuneration of medical procedures for the Supplementary Health) for the insertion and removal of contraceptives with a 4A rating, considering any additional examinations necessary for LNG-IUS insertion [37]. The prices of the therapeutic alternatives were retrieved from the 2021 drug price list of the Drug Market Regulation Chamber (*Câmara de Regulação do Mercado de Medicamentos*–CMED) through the factory price (PF), which is the value to be practiced by the manufacturing, importing, or distributing companies, and the maximum price allowed for sale to pharmacies, drugstores, and public administration entities [38]. These were adjusted by the purchasing power parity (PPP) of Brazil in 2021. Only the direct costs of the interventions were considered, and both costs and benefits were discounted at a rate of 5% according to the methodological guidelines of the Ministry of Health (*Ministério da Saúde*–MS) [36].

Based on the results of a decision tree analysis, two scenarios were proposed: LNG-IUS or ENG insertion. It was assumed that 100% of the target population would use the LNG-IUS modality—including the insertion and removal procedures, four consultations (two pre-insertion and one pre-removal), two intravaginal ultrasounds, and the cost of the device—at year zero of the analysis. Subsequently, the market share for the first year after incorporation was assumed to be 7% for the implant vs. 93% for LNG-IUS. This was followed by an assumed constant growth of 5% per year in subsequent years, as per data published by ANS, until the market share reached 27% for ENG and 73% LNG-IUS in the fifth year of the analysis. In accordance with real conditions in Brazil, market shares of 81.0% and 82.6% for LNG-IUS and ENG, respectively, were also accounted for in the calculations [39]. The costs of adverse events were not accounted for in regard to either method because a recent meta-analysis found no significant differences in safety between these techniques [5].

We performed a deterministic sensitivity analysis to identify the possible effects of fluctuations in the estimated values of the included variables. The upper and lower limits of each variable were set at 20% using the MS Excel program (Microsoft 365 MSO, version 2016). The variables were individually analyzed over a calculation period of five years and assessed according to their amplitude. Our results highlighted the overall cost of the procedures (including the cost of devices, procedures, and examinations), number of consultations, and market share. The market share was chosen as an outcome variable because it has the strongest impact on the budget.

## Results

Based on previous analytical studies and reviews, we gathered information on various aspects of IMP-ETN to evaluate how this contraceptive method can be offered across Brazil.

### Pharmacoeconomic analysis

In June 2021, the ANS oversaw regulation of 702 private healthcare providers with active beneficiaries. Of these providers, 20 with the highest numbers of beneficiaries accounted for 50.03% of all users, and 71 providers accounted for 70.18% of all users. The total number of beneficiaries was 47,768,176. A total of 53.09% (25,358,355) of beneficiaries were female, of whom 22.87% (5,799,229) were children or adolescents (aged 0–19 years), 43.39% (11,001,855) were adult and fertile women (20–44 years), and 33.75% (8,557,271) were mature or elderly women (45–80 years or more) [34]. To this number, we applied the percentage of general contraceptive use (84.8%, as identified by Lago et al). Our analysis revealed that a total of 9,329,573 beneficiaries in the Brazilian private healthcare system use contraception [35].

**Table 1. Summary of long-acting reversible contraceptive methods (LARC) used in Brazil.**

| Contraceptive method | Years of use according to approved label | Main characteristics | Non-contraceptive benefits |
|---|---|---|---|
| Tcu380A IUD | 10 | Not hormonal; pain and bleeding disturbances are the main reasons for discontinuation; | Reduction of likelihood of cervical and endometrial cancer |
| | | highly effective as an emergency contraception method | |
| LNG 52 mg IUS | 5 | Hormonal, estrogen-free; bleeding disturbances and hormonal side effects are the main reasons for discontinuation | Reduction of likelihood of endometriosis- associated pain, dysmenorrhea, and ovarian and endometrial cancer |
| LNG 19.5 mg IUS | 5 | Smaller device, hormonal, estrogen-free; bleeding disturbances and hormonal side effects are the main reasons for discontinuation | |
| ENG implant | 3 | Hormonal; bleeding disturbances and hormonal side effects are the main reasons for discontinuation | Reduction of likelihood of endometriosis-associated pain, dysmenorrhea, and adenomyosis |

Adapted from Table 1 of Bahamondes et al. [1], the last column of Table 2 of Curtis et al. [41], and the 5th row of Table 1 of Mohit et al. [42]. Long-acting reversible contraceptives-IUDs/IUS and implants: A review. *International Journal of Pharmaceutical Sciences Review and Research*, 68, 135–142.

We extrapolated the number of h-IUD insertions in 2020 (176,174 IUD insertions) to the calculated population currently using contraception. The results revealed that 1.89% of contraceptive users in the private healthcare system use hormonal LARCs (h-LARC). This value is close to the 2.5% identified by Lago et al. in a survey with a probability sample of 4,000 women living in São Paulo [35]. These findings also corroborate the findings of Leon et al., who reported that less than 10% of sexually active women of childbearing age in Latin America used LARCs [40]. We then multiplied the number of beneficiaries with the percentage of women who use h-LARC (1.89%), which identified 158,696 women as the target population for this analysis.

LARC methods are infrequently used not only in Brazil, but throughout Latin America. Nevertheless, the costs related to this procedure have a significant impact on healthcare payers [35,40]. Considering the costs related to the insertion and removal procedures (Table 1) and the number of women identified as the target population for this analysis, we estimate that private healthcare providers would have spent approximately $ 322.8 million on LARCs in 2020.

In our analysis, we considered the continuation rates of both methods from a clinical trial conducted by Modesto et al. [39]. We specifically looked at the continuation rate after 12 months, as shown in Table 2. We did not take into account the continuation rates for the second year, but we applied a 67.2% rate for the third year [43]. This analysis has a limitation. One weakness is that we assumed women who stopped using one method would switch to SARCs methods, which are not covered by Brazilian HMOs. During our analysis, we found that the difference in insertion costs between the ENG and LNG-IUS methods predicted potential savings of R$1,737.43 per patient. Furthermore, considering the integration of technology in the private healthcare system and the availability of ENG for all LARC users of childbearing age (the target population in this analysis), we recommend adopting the ENG method, which could result in a cost avoidance of $7,640,804.02 for payers of the private health system in the first year. Additionally, this would lead to a cumulative cost saving of $82,455,254.43 over a five-year period (Table 3).

In the time horizon of our analysis, the incremental cost across all years was negative. The variation increased from the first to the third year and plateaued in year 4 (Table 4). However, the ratio between the two scenarios remained negative because of the need for implant reinsertion after 36 months (the period of proven efficacy mentioned in the medication package

**Table 2. Comparison of costs related to LNG-IUS vs. ENG insertion and removal [37,44,45].**

| Procedure | Description | Amount | Continuation rate |
|---|---|---|---|
| LNG-IUS | (Three) pre-insertion consultation(s) | $267,10 | 81.0% |
| | Hormonal intrauterine device–insertion | $331,47 | |
| | (One) pre-removal consultation | $89,03 | |
| | Hormonal intrauterine device–removal | $331,47 | |
| | Transvaginal ultrasound (two examinations) | $247,97 | |
| | 18% state-level sales tax (VAT/ICMS) | $380,99 | |
| | **Total amount (1 beneficiary)** | **$1.648,03** | **$2.034,61** |
| ENG | (One) pre-insertion consultation | $89,03 | 82.6% |
| | Hormonal subdermal contraceptive implant–insertion | $331,47 | |
| | Hormonal subdermal contraceptive implant–removal | $331,47 | |
| | (One) pre-removal consultation | $89,03 | |
| | 18% state-level sales tax (VAT/ICMS) | $271,44 | |
| | **Total amount (1 beneficiary)** | **$1.112,45** | **$1.346,79** |

insert). Diedrich et al. reported that 67.2% of LARC users desire reinsertion of the implant and that those who continue using this contraceptive method consider it safe and effective [43]. Cost avoidance was highest in year 3, at which point both methods were within the efficacy period recommended in the medication package insert. In years 4 and 5 (during which the ENG is reinserted in patients who desire it), the amount of cost avoidance decreased, albeit with an efficiency gain across all years. These results reflect an incremental negative impact throughout the period of analysis, highlighting the favorable cost-effectiveness profile of the less expensive alternative.

To understand the impact of market share variation on the cost difference between LNG-IUS and ENG, we performed a sensitivity analysis by increasing or decreasing the variable estimate by 20%, such that the upper and lower limits were 120% and 80% of the base value, respectively (Table 5).

A 20% decrease in the market share of ENG would change the budget impact by approximately $ 15,04 million (budget impact vs. sensitivity analysis). This would reduce the costs of the system, albeit to a lesser amount than that in the main scenario (from—$82.455.254,434 to —$67.285.849,272) (Table 6).

Conversely, a 20% increase in the market share of ENG would further reduce the total cost of the system from—$82.455.254,434 to—$97.624.659,592 (Table 7). The dissemination of this information among physicians would enact changes among professionals and users of the private healthcare system, further strengthening the potential cumulative cost avoidance [43]. The savings accumulated by adopting this intervention may be incorporated into other healthcare strategies [46].

## Discussion

This study explored the implications of using etonogestrel implants as a LARC option. Here, we considered the various attributes of this system (as evidenced in the literature) and analyzed its value perception for a pharmacoeconomic study, demonstrating the feasibility of offering this contraceptive method in the Brazilian private healthcare system. This implant matches other available LARC options and provides considerable benefits to women by allowing them to exercise their choice in family planning.

**Table 3. Medical eligibility criteria (MEC) for the initiation of LARC methods.**

| Conditions for which at least one LARC method should not be used (MEC 4) or should not generally be used (MEC 3)* | | | | |
|---|---|---|---|---|
| **Condition** | **Category of medical eligibility criteria** | | | **Comments** |
| | **Copper-containing IUD** | **LNG-IUD** | **Implant** | |
| **Pregnancy** | 4 | 4 | NA | The use of an implant is not needed; no known harm to the woman, to the course of her pregnancy, or to the fetus occurs if an implant is inadvertently used during pregnancy |
| **Distorted uterine cavity incompatible with IUD placement** | 4 | 4 | NA | An anatomical abnormality that distorts the uterine cavity might preclude proper IUD placement |
| **Current pelvic inflammatory disease, gonococcal or chlamydial infection, or purulent cervicitis** | 4 | 4 | 1 | Insertion of an IUD might worsen the condition |
| **Postpartum or postabortion sepsis** | 4 | 4 | — | Insertion of an IUD might worsen the condition |
| **Persistent intrauterine gestational trophoblastic disease** | 4 | 4 | 1 | An IUD should not be inserted because of the risk of perforation, infection, or hemorrhage |
| **Cervical cancer** | 4 | 4 | 2 | Concerns exist about the increased risk of infection and bleeding at insertion; the IUD will probably have to be removed at the time of cancer treatment |
| **Endometrial cancer** | 4 | 4 | 1 | Concerns exist about the increased risk of infection, perforation, or bleeding at insertion; the IUD will probably have to be removed at the time of cancer treatment |
| **Unexplained vaginal bleeding (raising suspicion of a serious condition)** | 4 | 4 | 3 | If pregnancy or an underlying pathologic condition (e.g., pelvic cancer) is suspected, it must be evaluated and the category adjusted after evaluation; irregular bleeding patterns associated with the method used might mask symptoms of underlying pathologic conditions |
| **Current breast cancer** | 1 | 4 | 4 | Hormonal stimulation may worsen the condition |
| **History of breast cancer with no evidence of disease for 5 years** | 1 | 3 | 3 | — |
| **Complicated solid-organ transplantation** | 3 | 3 | 2 | Data on risks and benefits are limited in this population |
| **Systemic lupus erythematosus (with severe thrombocytopenia)** | 3 | 2 | 2 | Concern exists about an increased risk of bleeding |
| **Systemic lupus erythematosus (with positive or unknown antiphospholipid antibodies)** | 1 | 3 | 3 | Concern exists about an increased risk of both arterial and venous thrombosis |
| **Severe, decompensated cirrhosis** | 1 | 3 | 3 | Hormonal exposure may worsen the condition |
| **Hepatocellular adenoma or hepatic malignancy** | 1 | 3 | 3 | Hormonal exposure may worsen the condition |

Adapted from Table 1 of Bahamondes et al. [1], the last column of Table 2 of Curtis et al. [41], and the 5th row of Table 1 of Mohit et al. [42]. Same in other tables.

The last technology submission process on the 2022 ANS list brought together different stakeholders, including almost 350 contributions in the open public consultation round. Remarkably, 95% of the contributions addressed the urgency, need, and acquired gain for

**Table 4. Five-year budget impact analysis: Cost comparison between the current scenario (Scenario 1) and the proposed scenario (Scenario 2).**

| Scenarios | | Cost/patient ($) | Year 1 | Year 2 | Year 3 | Year 4 | Year 5 | Total |
|---|---|---|---|---|---|---|---|---|
| **Scenario 1** | ENG | $2.034,61 | $0 | $0 | $0 | $0 | $0 | |
| | LNG-IUS | $1.346,79 | $322.884,148 | $345.486,038 | $369.670,061 | $395.546,965 | $423.235,253 | |
| | **Total** | | **$322.884,148** | **$345.486,038** | **$369.670,061** | **$395.546,965** | **$423.235,253** | **$1.856.822,464** |
| **Scenario 2** | ENG | $2.034,61 | $14.961,086 | $27.442,907 | $41.598,873 | $67.656,054 | $94.083,801 | |
| | LNG-IUS | $1.346,79 | $300.282,257 | $304.027,713 | $306.826,150 | $308.526,633 | $308.961,734 | |
| | **Total** | | **$315.243,344** | **$331.470,620** | **$348.425,023** | **$376.182,687** | **$403.045,535** | **$1.774.367,209** |
| **Efficiency gain (scenario 1 –scenario 2)** | | | **- $7.640,804** | **- $14.015,418** | **- $21.245,037** | **- $19.364,278** | **- $20.189,717** | **- $82.455,255** |

**Table 5. Sensitivity analysis of the market shares of LARCs: LNG-IUS and ENG.**

| Market share | | Year 1 | Year 2 | Year 3 | Year 4 | Year 5 |
|---|---|---|---|---|---|---|
| LARC | | 100% | 100% | 100% | 100% | 100% |
| 1st scenario– 20% decrease in the market share of ENG | | | | | | |
| LNG-IUS | | 94% | 90% | 86% | 82% | 78% |
| ENG | | 6% | 10% | 14% | 18% | 22% |
| 2nd scenario– 20% increase in the market share of ENG | | | | | | |
| LNG-IUS | | 92% | 86% | 80% | 74% | 68% |
| ENG | | 8% | 14% | 20% | 26% | 32% |

users of contraceptive methods in Brazil with access to private healthcare. This high index demonstrates the meaning of treatments and procedures not covered by the National Health System–SUS. Limited access to healthcare is a worldwide problem. Economies in crisis need to cut spending, calling into question the democratic and egalitarian notion of health. A country that values the healthcare experience of its citizens and strives to improve their quality of life also reaps the benefits of these sustained investments. However, for this improvement to come into effect, this value must be shared among the different stakeholders: patients, doctors, healthcare providers, regulatory agencies, hospitals, and clinics. The varied goals of healthcare providers must be realigned to create a system that is value-driven rather than volume-driven. The aim and purpose of the healthcare system should be to deliver high quality service and best value to its customers.

Continuously reviewing data, making efforts to remove barriers to contraceptive access, and improving patient-focused care are key strategies for improving the service offered by the private healthcare industry [47]. Healthcare professionals can be trained to recognize their own biases, and this would help ensure that reproductive healthcare is provided in a patient-centered manner while respecting women's preferences and values. Private healthcare is expected to monitor and evaluate the quality of service provided by its affiliated professionals, and a survey can reveal the actual standard of care provided in real-world settings. However, this approach alone is insufficient for transforming the provision of contraceptive care. Performance measures are crucial for truly improving clinical services and health outcomes and conducting surveys more frequently and regularly may be beneficial for exploring customer expectations [48].

The possibility of an unplanned pregnancy can cause agony to women (and often their partners and family members). An unplanned pregnancy poses several threats (such as a decline in the economic situation of the prospective parents, family issues, and interruption of life goals), and such situations are marked by decision-making with long-term commitments and

**Table 6. Scenario 1: Budget impact of a 20% decrease in the proposed market share of ENG.**

| Year | Reference scenario | Proposed scenario | Budget impact |
|---|---|---|---|
| 1 | $322.884.147,664 | $316.334.887,173 | -$6.549.260,586 |
| 2 | $345.486.038,005 | $333.806.523,357 | -$11.679.514,711 |
| 3 | $369.670.060,570 | $352.174.147,664 | -$17.495.913,036 |
| 4 | $395.546.965,162 | $380.095.173,001 | -$15.451.791,785 |
| 5 | $423.235.252,573 | $407.125.883,215 | -$16.109.369,157 |
| | | Efficiency gain | -$67.285.849,272 |

**Table 7. Scenario 2: Budget impact of a 20% increase in the market share of ENG.**

| Year | Reference Scenario | Proposed Scenario | Budget Impact |
|---|---|---|---|
| 1 | $322.884.147,664 | $314.151.800,08 | -$8.732.347,45 |
| 2 | $345.486.038,005 | $329.134.717,34 | -$16.351.320,59 |
| 3 | $369.670.060,570 | $344.675.899,05 | -$24.994.161,48 |
| 4 | $395.546.965,162 | $372.270.200,71 | -$23.276.764,32 |
| 5 | $423.235.252,573 | $398.965.186,86 | -$24.270.065,75 |
| | | Efficiency Gain | -$97.624.659,59 |

responsibility. Ferreira et al. (2014) reported that 1,154 women switched from a short-acting reversible contraceptive method to a LARC based on "the fear of becoming pregnant." Moreover, the authors indicated that the preferred contraceptive method was ENG-IMP when available; this may explain why the maintenance rate for the three LARC methods surpassed 90/100 women/year at the end of the first year after placement [28].

We believe that our analysis overcomes previous barriers to the use of cost-benefit models and produces desirable outcomes. Therefore, our results can serve as a basis upon which policymakers, technical consultants, and researchers can make decisions. This study provides a resource for healthcare providers who can help develop better family planning strategies and improve the reproductive health of women and their families. Contraceptive counseling should focus on shared decision-making models. Healthcare providers offer technical knowledge, whereas the patient provides information regarding their own priorities, values, and preferences [49]. However, sometimes physicians can also assume a more proactive position in the decision-making process by identifying conditions in which a subdermal implant would be suitable. These conditions can be direct or indirect indications that provide opportunities for the use of implants (depending on the characteristics of the product) (Table 8).

There is a lack of economic analysis regarding contraceptives, especially in developing countries. This poses a major challenge to reproductive healthcare, as the results of economic evaluations in developed countries may not be transferable to developing countries due to vastly different payment models (payer versus out-of-pocket) and contraceptive use patterns, among other factors [50]. Few studies have evaluated the economic parameters associated with implant use in a country with sanitary and economic characteristics similar to those in Brazil. One such study was conducted in India; the authors performed probabilistic sensitivity analysis and found that the addition of the ENG implant would be cost-effective 100% of the time and would represent <0.5% of the country's annual health budget [51].

Fig 6 shows the estimated disability-adjusted life years (DALY) of maternal morbidity and mortality, child mortality, total mortality, and averted unsafe abortions [52]. Although this DALY analysis was based on a mathematical model, we conclude that the use of LARCs can help fulfil the healthcare requirements of the most vulnerable segment of the population. Hence, programs offering different types of LARCs to the general population can progressively improve the DALY estimates of women. However, the gains were not equal across the population, and health inequality in this instance, as in general, remains a major challenge in Brazil.

The use of implants is associated with a very favorable Pearl index. Ferreira-Filho et al. discussed the safety and importance of this method and reported its high efficacy in women with Hansen's disease who were using thalidomide [53]. Krajewski et al. also supported these findings in a study on women with a history of solid-organ transplantation [54]. Machado et al. discussed the clinical conditions represented by different categories of Medical Eligibility Criteria for contraceptive use in Brazil [27]; based on their findings, the authors reported that implants are recommended under specific conditions and have more indications than other

**Table 8. Patient groups for whom the subdermal IMP-ETN implant may be suitable, and conditions indicating suitability.**

| Groups for which subdermal implants may be suitable | Plausibility |
|---|---|
| **Adolescents** | May show better acceptance as a long-acting reversible method that does not require any action from the patient, it is not externally visible, but still palpable, it spares the need for a special appointment and reexams, guaranteeing efficacy with convenience. |
| **Postpartum** | Women may experience sensitivity during this period. A same-day insertion method that does not affect expulsion rates could be a plus. |
| **Adult women with specific contraindications to available methods** | Situations like the ones below can occur in a large population:<br>• Congenital or acquired uterine anomaly (including leiomyomas) that cause deformation of the uterine cavity;<br>• Current or recurrent pelvic inflammatory disease;<br>• Lower genital tract infection;<br>• Postpartum endometritis;<br>• Infected abortion during the last three months;<br>• Cervicitis;<br>• Cervical dysplasia;<br>• Uterine or cervical malignant tumor;<br>• Conditions associated with increased susceptibility to infections;<br>• History of arterial or venous thrombotic/thromboembolic processes (e.g., deep vein thrombosis, pulmonary embolism, or myocardial infarction); or a stroke;<br>• History of migraine with focal neurological symptoms;<br>• Diabetes mellitus with vascular changes. |
| **Women who have not adapted to the use of intrauterine devices** | Women who have previously tried intrauterine devices or systems may not have adapted to them. LARCs are the most efficient contraception method, and subdermal implants may be the correct option. |

Prepared by the authors of this study.

LARCs. Wigginton et al. discussed various reasons (such as pain) that may cause women to change their preferred contraceptive method [55].

Implants are highly effective as contraceptives, showing the highest efficacy (~99.95%) of all available methods. The body mass index of users does not seem to interfere with its efficacy. In Brazil, the etonogestrel-releasing implant is the only implant approved by the ANVISA for 3 years of use. The most common side effect of implants is unpredictable bleeding, and investigators generally recommend that women be counseled about this risk before implant placement [41]. Although the impact and success of healthcare policies also depends on various social determinants [26,55], these indicators are often not enough to determine of the effectiveness of the services provided. Therefore, healthcare providers should pay special attention to the principles of care by ensuring the improvement and adequacy of care systems and expanding treatment and follow-up techniques.

This study has two main limitations. First, we only considered the direct costs involved in the two contraceptive methods and did not consider indirect costs. In addition, there is a lack of Brazilian reference data on contraceptive methods in an environment of free choice without barriers to accessibility. This makes it difficult to accurately determine the proposed market share of each method.

## Conclusion

In conclusion, we provide a comprehensive view of the feasibility and reliability of the etonogestrel implant technology and explore its beneficial impacts on women's quality of life.

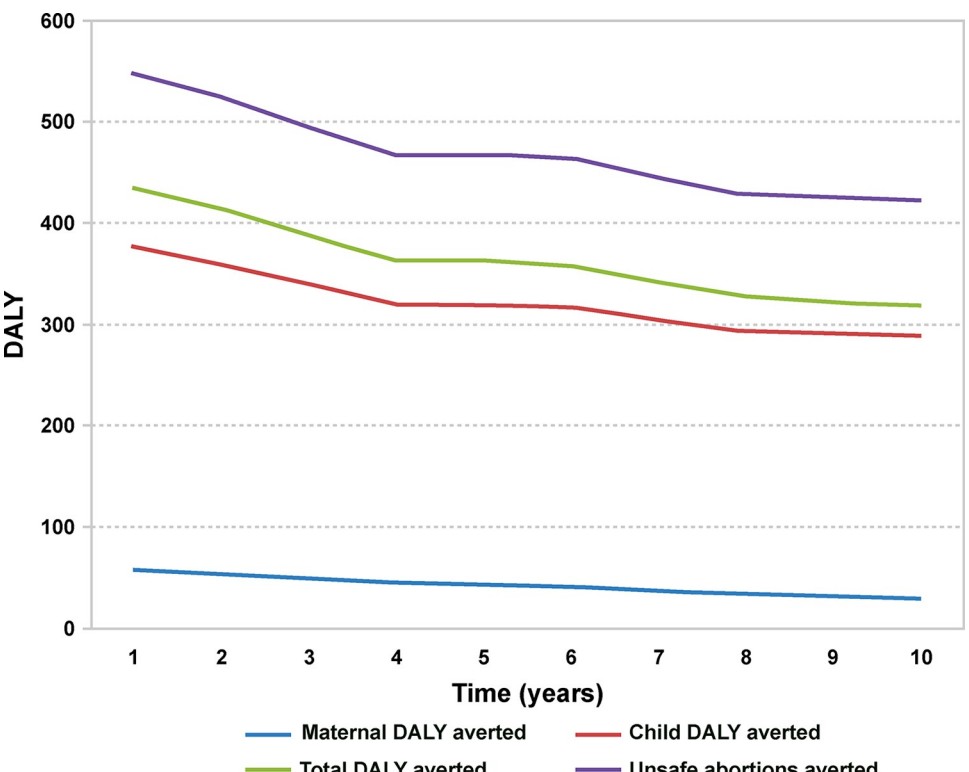

**Fig 6. Estimated disability-adjusted life years (DALY) of indicators of maternal and child health.** Created by the authors—Adapted from [52].

LARCs are an increasingly popular contraceptive option for women in any stage of their reproductive lifespan, and the method discussed here is convenient, effective, and reversible. The data presented here lead us to strongly recommend that policy makers, decision makers, and health professionals come together to promote LARCs technologies as a cost-effective method of contraception. It is implied that we have not yet obtained results on the validity of the initiative, as our work only considered hypothetical scenarios. A validity test would be necessary to determine the extent of interest from decision makers and to establish the credibility of the gains, value, threats, and challenges of each LARC.

Each contraceptive method is unique and offers specific benefits to users. However, these methods have not yet been fully explored by private healthcare providers in Brazil. As such, users are compelled into a restrictive "one-size-fits-all" model. Our literature search revealed studies that highlight the practicality, ease of handling, consistent efficacy, and general safety of etonogestrel subdermal implants. Our analysis suggests that the therapeutic alternative presented here had a favorable cost-effectiveness profile and showed incremental negative impacts throughout the period of analysis. This analytical model allowed us to identify a cost gain of $ 687,82 per beneficiary who is a user of LARCs (compared with users of LNG-IUS) in the Brazilian private healthcare system. Over a five-year time horizon, ENG was found to be a less expensive alternative than LNG-IUS, even when factoring in the reinsertion costs.

Given the aforementioned advantages of this system, ENG emerges as an alternative contraceptive choice with a more favorable cost-effectiveness profile and lower budgetary impact than LNG-IUS. Therefore, it should be offered to beneficiaries with a private health plan. Expanding the offer of LARC methods to users of the Brazilian private healthcare system

guarantees the right of doctors and patients to choose the therapeutic alternative with the greatest benefit while simultaneously promoting sustainable market competition.

## Author Contributions

**Conceptualization:** Agnaldo Lopes da Silva Filho, Ricardo Luis Pereira Bueno.

**Data curation:** Ricardo Luis Pereira Bueno, Yohanna Ramires.

**Formal analysis:** Yohanna Ramires, Lara Marina Cruz Lino.

**Funding acquisition:** Ricardo Luis Pereira Bueno, Yohanna Ramires, Lara Marina Cruz Lino.

**Investigation:** Ricardo Luis Pereira Bueno, Yohanna Ramires, Lara Marina Cruz Lino.

**Methodology:** Ricardo Luis Pereira Bueno, Yohanna Ramires, Lara Marina Cruz Lino.

**Project administration:** Ricardo Luis Pereira Bueno, Yohanna Ramires, Lara Marina Cruz Lino.

**Resources:** Yohanna Ramires.

**Supervision:** Agnaldo Lopes da Silva Filho, Ricardo Luis Pereira Bueno.

**Validation:** Agnaldo Lopes da Silva Filho, Ricardo Luis Pereira Bueno, Yohanna Ramires.

**Visualization:** Agnaldo Lopes da Silva Filho, Ricardo Luis Pereira Bueno, Yohanna Ramires, Lara Marina Cruz Lino.

**Writing – original draft:** Agnaldo Lopes da Silva Filho, Yohanna Ramires, Lara Marina Cruz Lino.

**Writing – review & editing:** Agnaldo Lopes da Silva Filho, Ricardo Luis Pereira Bueno, Yohanna Ramires, Lara Marina Cruz Lino.

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
