## [Decision Letter · Decision Letter 0]

21 Jun 2023

PONE-D-23-06711Etonogestrel-releasing subdermal contraceptive implant: Budget impact analysis based on the Brazilian private healthcare systemPLOS ONE

Dear Dr. Lino,

Thank you for submitting your manuscript to PLOS ONE. After careful consideration, we feel that it has merit but does not fully meet PLOS ONE’s publication criteria as it currently stands. Therefore, we invite you to submit a revised version of the manuscript that addresses the points raised during the review process.

Please respond to all reviewers comments point by point============================================================

We look forward to receiving your revised manuscript.

Kind regards,

Ahmed Mohamed Maged, MD

Academic Editor

PLOS ONE

Journal Requirements:

When submitting your revision, we need you to address these additional requirements. 1. Please ensure that your manuscript meets PLOS ONE's style requirements, including those for file naming. The PLOS ONE style templates can be found at https://journals.plos.org/plosone/s/file?id=wjVg/PLOSOne_formatting_sample_main_body.pdf and https://journals.plos.org/plosone/s/file?id=ba62/PLOSOne_formatting_sample_title_authors_affiliations.pdf 2. We note that the grant information you provided in the ‘Funding Information’ and ‘Financial Disclosure’ sections do not match.  When you resubmit, please ensure that you provide the correct grant numbers for the awards you received for your study in the ‘Funding Information’ section. 3. Thank you for stating the following financial disclosure: "The authors received no specific funding for this work. ⦁
Initials of the authors who received each award - THERE WERE NO AWARD⦁
Grant numbers awarded to each author - THERE WERE NO GRANT⦁
The full name of each funder - ORGANON BRASIL⦁
URL of each funder website - ORGANON.COM.BR⦁
Did the sponsors or funders play any role in the study design, data collection and analysis, decision to publish, or preparation of the manuscript? LARA LINO, YOHANNA RAMIRES, RICARDO BUENO ALL OF THE ABOVE - WE THREE DECIDED TO PUBLISH, COLLECTED THE DATA, WROTE AND INVITED AGNALDO LOPES TO REVISE AND INPUTS." At this time, please address the following queries: a) Please clarify the sources of funding (financial or material support) for your study. List the grants or organizations that supported your study, including funding received from your institution. b) State what role the funders took in the study. If the funders had no role in your study, please state: “The funders had no role in study design, data collection and analysis, decision to publish, or preparation of the manuscript.”c) If any authors received a salary from any of your funders, please state which authors and which funders.d) If you did not receive any funding for this study, please state: “The authors received no specific funding for this work.” Please include your amended statements within your cover letter; we will change the online submission form on your behalf. 4. Thank you for stating the following in the Competing Interests section: "I have read the journal's policy and the authors of this manuscript have the following competing interests: Ricardo Luis Pereira Bueno, Yohanna Ramirez and Lara Marina Cruz Lino are employees of Organon Brasil [Organization that funded the study]." We note that you received funding from a commercial source: ORGANON BRASILPlease provide an amended Competing Interests Statement that explicitly states this commercial funder, along with any other relevant declarations relating to employment, consultancy, patents, products in development, marketed products, etc.  Within this Competing Interests Statement, please confirm that this does not alter your adherence to all PLOS ONE policies on sharing data and materials by including the following statement: ""This does not alter our adherence to PLOS ONE policies on sharing data and materials.” (as detailed online in our guide for authors http://journals.plos.org/plosone/s/competing-interests).  If there are restrictions on sharing of data and/or materials, please state these. Please note that we cannot proceed with consideration of your article until this information has been declared.  Please include your amended Competing Interests Statement within your cover letter. We will change the online submission form on your behalf. 5. In your Data Availability statement, you have not specified where the minimal data set underlying the results described in your manuscript can be found. PLOS defines a study's minimal data set as the underlying data used to reach the conclusions drawn in the manuscript and any additional data required to replicate the reported study findings in their entirety. All PLOS journals require that the minimal data set be made fully available. For more information about our data policy, please see http://journals.plos.org/plosone/s/data-availability. Upon re-submitting your revised manuscript, please upload your study’s minimal underlying data set as either Supporting Information files or to a stable, public repository and include the relevant URLs, DOIs, or accession numbers within your revised cover letter. For a list of acceptable repositories, please see http://journals.plos.org/plosone/s/data-availability#loc-recommended-repositories. Any potentially identifying patient information must be fully anonymized. Important: If there are ethical or legal restrictions to sharing your data publicly, please explain these restrictions in detail. Please see our guidelines for more information on what we consider unacceptable restrictions to publicly sharing data: http://journals.plos.org/plosone/s/data-availability#loc-unacceptable-data-access-restrictions. Note that it is not acceptable for the authors to be the sole named individuals responsible for ensuring data access. We will update your Data Availability statement to reflect the information you provide in your cover letter. 6. PLOS requires an ORCID iD for the corresponding author in Editorial Manager on papers submitted after December 6th, 2016. Please ensure that you have an ORCID iD and that it is validated in Editorial Manager. To do this, go to ‘Update my Information’ (in the upper left-hand corner of the main menu), and click on the Fetch/Validate link next to the ORCID field. This will take you to the ORCID site and allow you to create a new iD or authenticate a pre-existing iD in Editorial Manager. Please see the following video for instructions on linking an ORCID iD to your Editorial Manager account: https://www.youtube.com/watch?v=_xcclfuvtxQ 7. We note that Figures 1, 2 and 4 in your submission contain copyrighted images. All PLOS content is published under the Creative Commons Attribution License (CC BY 4.0), which means that the manuscript, images, and Supporting Information files will be freely available online, and any third party is permitted to access, download, copy, distribute, and use these materials in any way, even commercially, with proper attribution. For more information, see our copyright guidelines: http://journals.plos.org/plosone/s/licenses-and-copyright. We require you to either (1) present written permission from the copyright holder to publish these figures specifically under the CC BY 4.0 license, or (2) remove the figures from your submission: a. You may seek permission from the original copyright holder of Figures 1, 2 and 4 to publish the content specifically under the CC BY 4.0 license.  We recommend that you contact the original copyright holder with the Content Permission Form (http://journals.plos.org/plosone/s/file?id=7c09/content-permission-form.pdf) and the following text:“I request permission for the open-access journal PLOS ONE to publish XXX under the Creative Commons Attribution License (CCAL) CC BY 4.0 (http://creativecommons.org/licenses/by/4.0/). Please be aware that this license allows unrestricted use and distribution, even commercially, by third parties. Please reply and provide explicit written permission to publish XXX under a CC BY license and complete the attached form.” Please upload the completed Content Permission Form or other proof of granted permissions as an ""Other"" file with your submission.  In the figure caption of the copyrighted figure, please include the following text: “Reprinted from [ref] under a CC BY license, with permission from [name of publisher], original copyright [original copyright year].” b. If you are unable to obtain permission from the original copyright holder to publish these figures under the CC BY 4.0 license or if the copyright holder’s requirements are incompatible with the CC BY 4.0 license, please either i) remove the figure or ii) supply a replacement figure that complies with the CC BY 4.0 license. Please check copyright information on all replacement figures and update the figure caption with source information. If applicable, please specify in the figure caption text when a figure is similar but not identical to the original image and is therefore for illustrative purposes only.

Additional Editor Comments:

Please respond to all reviewers comments

Reviewers' comments:

Reviewer's Responses to Questions

**Comments to the Author**

1. Is the manuscript technically sound, and do the data support the conclusions?

Reviewer #1: Yes

Reviewer #2: Yes

2. Has the statistical analysis been performed appropriately and rigorously? 

Reviewer #1: I Don't Know

Reviewer #2: Yes

3. Have the authors made all data underlying the findings in their manuscript fully available?

Reviewer #1: No

Reviewer #2: Yes

4. Is the manuscript presented in an intelligible fashion and written in standard English?

Reviewer #1: Yes

Reviewer #2: Yes

5. Review Comments to the Author

Reviewer #1: Thank you for asking me to review the manuscript titled “ETONOGESTREL-RELEASING SUBDERMAL CONTRACEPTIVE IMPLANT: BUDGET IMPACT ANALYSIS BASED ON THE BRAZILIAN PRIVATE HEALTHCARE SYSTEM”. (PONE-D-23-06711)

I have gone through the manuscript and the following are my observations and comments:

1. It is not clear what the study set out to address as the reference to “cost avoidance” did not state who the cost avoidance relates to.

2. Cost alone, as the Authors also pointed out, is not the only consideration for contraceptive choice.

3. No human subjects were used to test the validity of all the positive and negative impacts the Authors attributed to the contraceptive (IMP-ETN) and references to published articles were limited.

4. The article appears more like a “marketing strategy” presentation with obvious bias towards the product IMP-ETN and therefore does not qualify for publication in PLOS ONE. This is not surprising judging from the fact that some of the Authors have conflicting interests being employees of the sponsoring organisation (ORGANON).

Reviewer #2: This article is an economic analysis that assesses the cost-effectiveness of adopting IMP-ETN over a period of five year compared to LNG-IUS. The author has the strength of presenting evidence in a setting that previous evidence is scarce – few economic analysis has been performed on this topic in the Brazilian setting, and therefore this article would be interesting to readers working in the country and neighbouring region, or even countries that are socio-economically similar. However revisions are needed as charted below.

As this study derives the outcome figures based on available data, the conclusion is only as good as the data. Therefore more justification on the validity of the data sets, the references, and the figures selected should be added in the methodology section. For example in line 131, it would be easier for readers to understand the methodology by explaining what specific data are retrieved from the ANS, and how comprehensive and accurate are the data on ANS (e.g. any data checks by ANS, any previous study validating data of ANS). It would also be good to explain if the costs defined by CBHPM (line 150-151) and CMED (line 154-155) correlate to the cost actually imposed by the fee-for-service private healthcare sector – as many readers are unfamiliar with the Brazilian medical system, one wonders if the costs in these lists are legally binding or whether private sectors can inflate the price as they wish. On the other hand, why is ANS data on the actual cost for procedures not used, is it because such data is not available in ANS?

I am also a bit confused by line 147, which mentions an assumed annual growth of 7% - does this refer to the growth in target population, or growth of IMP-ETN market share? Better clarify that. Also for line 165, is there any justification for the assumed constant growth of 5% per year? I understand it may be difficult to accurately make estimations, and the authors have also acknowledged this issue in the discussion.

Another suggestion I have is to move the literature review (line 186-259) to the introduction or discussion, as the main outcome should be the on the cost-effectiveness analysis.

6. PLOS authors have the option to publish the peer review history of their article (what does this mean?). If published, this will include your full peer review and any attached files.

Reviewer #1: **Yes: **Dr Francis E. Alu

Consultant Obstetrician & Gynaecologist

Abuja Nigeria

Reviewer #2: **Yes: **Liu Qinyang

---

## [Author Response · Author response to Decision Letter 0]

29 Jan 2024

Response to Reviewers file uploaded in the Attach Files section.

---

## [Decision Letter · Decision Letter 1]

13 Mar 2024

Etonogestrel-releasing subdermal contraceptive implant: budget impact analysis based on the Brazilian private healthcare system

PONE-D-23-06711R1

Dear Dr. Lino,

We’re pleased to inform you that your manuscript has been judged scientifically suitable for publication and will be formally accepted for publication once it meets all outstanding technical requirements.

Kind regards,

Ahmed Mohamed Maged, MD

Academic Editor

PLOS ONE

Additional Editor Comments (optional):

Reviewers' comments:

Reviewer's Responses to Questions

**Comments to the Author**

1. If the authors have adequately addressed your comments raised in a previous round of review and you feel that this manuscript is now acceptable for publication, you may indicate that here to bypass the “Comments to the Author” section, enter your conflict of interest statement in the “Confidential to Editor” section, and submit your "Accept" recommendation.

Reviewer #1: All comments have been addressed

2. Is the manuscript technically sound, and do the data support the conclusions?

Reviewer #1: Yes

3. Has the statistical analysis been performed appropriately and rigorously? 

Reviewer #1: Yes

4. Have the authors made all data underlying the findings in their manuscript fully available?

Reviewer #1: Yes

5. Is the manuscript presented in an intelligible fashion and written in standard English?

Reviewer #1: Yes

6. Review Comments to the Author

Reviewer #1: The Authors have substantially addressed the issues i raised. The explanation as to what the cost implication is referring to has been addressed as has other comments relating to comparison with other forms of LARCs. The economic analysis of the study and the cost-effectiveness of IMP-ETN in the country of study have been elucidated. i recommend its acceptance for publication as amended.

7. PLOS authors have the option to publish the peer review history of their article (what does this mean?). If published, this will include your full peer review and any attached files.

Reviewer #1: No

---

## [Editor Report · Acceptance letter]

18 Mar 2024

PONE-D-23-06711R1 

PLOS ONE

Dear Dr. Lino, 

I'm pleased to inform you that your manuscript has been deemed suitable for publication in PLOS ONE. Congratulations! Your manuscript is now being handed over to our production team.

Kind regards, 

on behalf of

Professor Ahmed Mohamed Maged 

Academic Editor

PLOS ONE